# Dynamics of growing carbon nanotube interfaces probed by machine learning-enabled molecular simulations

Daniel Hedman [1] ✉, Ben McLean [1,2], Christophe Bichara [3],
Shigeo Maruyama [4], J. Andreas Larsson [5] ✉ & Feng Ding [1,6,7] ✉

Carbon nanotubes (CNTs), hollow cylinders of carbon, hold great promise for advanced technologies, provided their structure remains uniform throughout their length. Their growth takes place at high temperatures across a tube-catalyst interface. Structural defects formed during growth alter CNT properties. These defects are believed to form and heal at the tube-catalyst interface but an understanding of these mechanisms at the atomic-level is lacking. Here we present DeepCNT-22, a machine learning force field (MLFF) to drive molecular dynamics simulations through which we unveil the mechanisms of CNT formation, from nucleation to growth including defect formation and healing. We find the tube-catalyst interface to be highly dynamic, with large fluctuations in the chiral structure of the CNT-edge. This does not support continuous spiral growth as a general mechanism, instead, at these growth conditions, the growing tube edge exhibits significant configurational entropy. We demonstrate that defects form stochastically at the tube-catalyst interface, but under low growth rates and high temperatures, these heal before becoming incorporated in the tube wall, allowing CNTs to grow defect-free to seemingly unlimited lengths. These insights, not readily available through experiments, demonstrate the remarkable power of MLFF-driven simulations and fill long-standing gaps in our understanding of CNT growth mechanisms.

Carbon nanotubes (CNTs) stand as an iconic example of low-dimensional materials. These hollow tubes, composed of carbon atoms arranged in a hexagonal lattice[1], have diameters of only a few nanometers yet can extend several centimeters in length[2–4]. Over the past three decades, researchers have discovered remarkable mechanical[5], thermal[6], electrical[7], and optical[8] properties of CNTs. Their electrical properties can be precisely

tailored by adjusting the orientation of the hexagonal lattice relative to the tube axis[9], represented by two chiral indices ($n,m$), making CNTs highly attractive for advanced technologies[10–13]. However, maintaining uniform properties over their entire length is challenging, as the chirality must be constant along the length of the tube. Changes in chirality result from defects in the tube wall, typically in the form of pentagons or heptagons which form

[1]Center for Multidimensional Carbon Materials (CMCM), Institute for Basic Science (IBS), Ulsan 44919, Republic of Korea. [2]School of Engineering, RMIT University, Victoria 3001, Australia. [3]Aix-Marseille Univ, CNRS, CINaM, UMR7325, Marseille 13288, France. [4]Department of Mechanical Engineering, The University of Tokyo, Tokyo 113-8656, Japan. [5]Applied Physics, Division of Materials Science, Department of Engineering Sciences and Mathematics, Luleå University of Technology, Luleå 971 87, Sweden. [6]Department of Materials Science and Engineering, Ulsan National Institute of Science and Technology (UNIST), Ulsan 44919, Republic of Korea. [7]Faculty of Materials Science and Engineering, Institute of Technology for Carbon Neutrality, Shenzhen Institute of Advanced Technology Chinese Academy of Sciences, Shenzhen 518055, China. ✉e-mail: daniel.hedman@ltu.se; andreas.1.larsson@ltu.se; f.ding@siat.ac.cn

during synthesis[14]. A typical centimeter-long single-walled carbon nanotube (SWCNT) consists of approximately $10^{10}$ hexagons; thus the defect concentration must be less than 0.1 parts per million to produce long defect-free SWCNTs.

Catalytic chemical vapor deposition has emerged as the most prominent method for synthesizing CNTs, employing metal nano-particles as catalysts to decompose hydrocarbon gas at high temperatures[15]. Among the metals, iron is one of the most widely used and resides in the middle of the Goldilocks' zone of metals considered as effective catalysts for synthesizing CNTs[16,17]. From these decom-posed hydrocarbons, an initial CNT-cap nucleates on the catalyst from carbon monomers and dimers. If the thermodynamic driving force is large enough, the cap will lift off the catalyst and form the tip of the developing CNT[18], which elongates (grows) through continuous incorporation of carbon atoms at the interface between the CNT-edge and the catalyst (the tube-catalyst interface). For SWCNTs, the rate of carbon incorporation (growth rate) spans from 0.5 to 10 carbon atoms per microsecond[2,3,15,19,20]. Fundamental understanding of the mechan-isms behind CNT nucleation and growth, i.e. the evolution of the tube-catalyst interface, is crucial for producing long defect-free CNTs with uniform properties throughout their length. While experimental stu-dies, particularly in situ transmission electron microscopy, have pro-vided valuable insights[21–23], a comprehensive atomic-level understanding of CNT growth has not yet been achieved through experimental measurements alone. Instead, computational studies, especially molecular dynamics (MD), have played a crucial role in revealing aspects of the growth mechanisms[24]. However, MD simula-tions have been methodologically limited in accurately exploring the timescales necessary for defect-free growth without the use of addi-tional biasing methods[25,26]. Consequently, the growth of defect-free CNTs by unbiased MD simulations remains elusive[27], and many ques-tions related to growth remain unanswered. Namely, the timescale of the nucleation process, how defects form and heal, and the evolution of the tube-catalyst interface during growth—all of which are crucial to understand in order to achieve controlled growth of long defect-free CNTs.

In this work, we develop DeepCNT-22, a machine learning force field[28] (MLFF) based on DeePMD[29] and use it to drive near-microsecond timescale MD simulations of SWCNT growth on iron catalysts. DeepCNT-22 enables us to investigate the entire SWCNT growth pro-cess without sacrificing computational accuracy and without employing steering or other biases. We reveal the timescales of nucleation and the mechanisms of growth, including the evolution of the tube-catalyst interface which we found exhibits significant con-figurational entropy. Achieving defect-free growth allows us to study defect formation and healing at the tube-catalyst interface, which was found to rely on the interplay between growth rate and temperature. This work represents significant progress in the theoretical under-standing of SWCNT growth and can be leveraged to guide catalytic chemical vapor deposition for controlled growth.

## Results

MLFFs are an emerging and powerful method for modeling materials at length and timescales that approach experiment. This method involves training machine learning models on a large dataset of atomic configurations (structures) labeled with energies, forces and virials calculated using first principles methods such as density functional theory (DFT). Once trained, MLFFs can predict physical quantities and drive atomistic simulations with the computational efficiency of empirical force fields, all while maintaining the accuracy of DFT or even beyond-DFT methods[30].

A significant challenge when developing MLFFs is creating high-quality, diverse datasets for training. The DeepCNT-22 dataset[31], shown in Fig. 1 as a sketch-map representation[32], includes a wide variety of structures relevant to SWCNT growth. Each point in the sketch-map denotes a unique structure, with its position determined by principal component analysis of the learned descriptors of the local atomic environments.

As the atomic configurations illustrate, different regions of the sketch-map correspond to different structures, with clear grouping of similar structures and separation of dissimilar ones. This highlights the diversity of the dataset and the quality of the learned descriptors. Details on the creation of this dataset as well as training of the DeepCNT-22 MLFF are provided in the Methods section. Verification of the accuracy of DeepCNT-22—including its ability to accurately reproduce the expected broad chirality distribution typically found for iron catalysts and the ratio of SWCNT diameter to catalyst diameter—is provided in Section 1 of the Supplementary Information. In addition, the accuracy of the MLFF was continuously monitored during the MD simulations via model deviation[33], which, as seen in Fig. S1c, has a single peak centered around 250 meV $Å^{-1}$. Thus, accuracy is maintained throughout the growth simulation with no bias.

After training, DeepCNT-22 was used to drive MD simulations of SWCNT growth starting from clean iron catalysts. In these simula-tions, the carbon supply rate, $k$, and the growth temperature, $T$, are parameters that influence the growth process. A carbon supply rate of $k \leq 1.0$ $ns^{-1}$ matched with a growth temperature of $1200 \leq T \leq 1500$ K is found to be suitable for growth, as seen in Fig. S9. Under these conditions, the growth rate is limited by the carbon supply rate, resulting in a 1:1 correlation between them, thus both terms are used interchangeably. Figure 2 shows the result of a 4.76 nm long $(n,m) = (6,5)$ SWCNT grown on a $Fe_{55}$ catalyst over 0.852 μs at $T = 1300$ K and $k = 0.5$ $ns^{-1}$. This corresponds to a growth rate of 5590 μm $s^{-1}$, which is approximately 50 to 1000 times higher than experimentally reported growth rates[2,3,15,19,20] and lower by a factor of up to 100 compared to previous MD simulations[26,27,34–36]. Despite the high growth rate, the resulting SWCNT shown in Fig. 2a is free of defects, demonstrating that defect-free growth can be achieved even at high growth rates. Additional defect-free SWCNTs grown using DeepCNT-22 can be found in Fig. S10. It should be noted that the chirality of these tubes is not predetermined but emerges naturally during the growth simulation.

As marked by the vertical lines in Fig. 2b, growth can be divided into five distinct phases. 1st abundance of carbon monomers inside the catalyst and dimers on the surface, 2nd conversion of monomers and dimers into carbon chains, 3rd rapid conversion of chains into gra-phitic carbon (pentagons and hexagons), 4th formation of the SWCNT-cap and cap liftoff, and 5th continuous elongation of the tube. Though these five phases have in part been investigated in previous studies[25,26,34,37,38], here the entire process is presented in full and unveils the timescale of each phase. A detailed breakdown of which can be found in Section 2 of the Supplementary Information. These five phases, combined with the snapshots in Fig. 2a and Supplementary Movie 1, offer comprehensive atomic-level details of SWCNT nuclea-tion and growth.

Note that here carbon atoms are supplied directly inside the catalyst, which then diffuse rapidly to the surface, rather than via hydrocarbon ($CH_x$) decomposition. Previous MD simulations[39] have shown that $CH_x$ undergoes C-H bond cleavage on the catalyst depos-iting both carbon and hydrogen on the surface, a process with low energy barriers[40]. However, it has been shown that these surface-bound hydrogens are few in number and not present at the tube-catalyst interface[41]. In addition, DFT calculations presented in Section 5 of the Supplementary Information show that adsorbed hydrogen on the surface of the catalyst, see Supplementary Data 1, only marginally increases the carbon-metal adhesion energy. Thus, while the presence of hydrogen may passivate dangling carbon bonds and slow down nucleation in the early growth stages, most hydrogen eventually des-orbs from the surface and those that remain do not significantly affect the Fe-C bond strength at the interface.

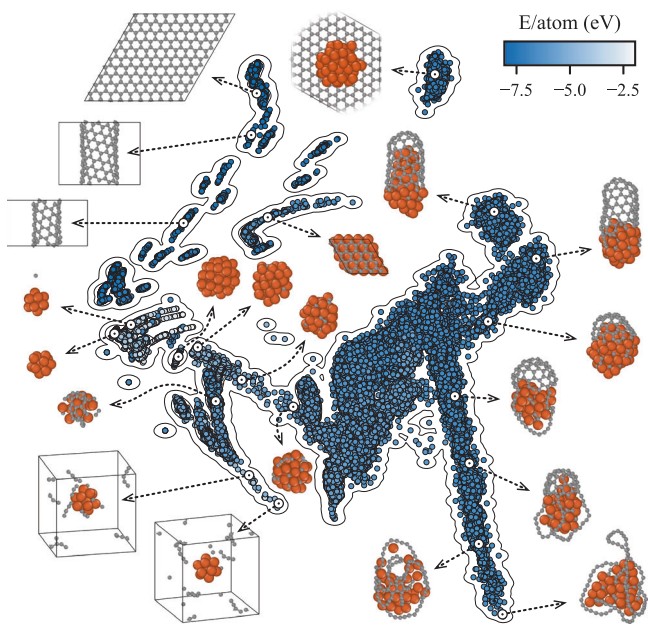

**Fig. 1 | Sketch-map visualization of the DeepCNT-22 dataset.** A sketch-map consisting of 22,975 structures where each colored dot represents an individual atomic configuration (structure). The position of each dot is determined by principal component analysis of the learned descriptors of the structures and its color indicates the corresponding energy of the structure. Examples of atomic configurations from different regions of the sketch map are shown to provide insight into the diversity of the data set. Here orange and grey spheres represent Fe and C atoms, respectively. Source data for this figure is provided in the Source Data file.

## Defect formation and healing

As seen from the snapshot at $t = 852.00$ ns in Fig. 2a, the grown SWCNT is straight and of single chirality, which is only possible if the tube wall consists solely of hexagons. However, this does not mean that only hexagons are formed during growth. Analysis of the number of penta-, hexa-, and heptagons during the 5th phase of growth, to the right of the dashed line in Fig. 2c, reveals a continuous increase in the number of hexagons where the rate of hexagon formation $k_6$ is half the growth rate, i.e., $k_6 = \frac{k}{2} = 0.25$ ns$^{-1}$. Moreover, the number of pentagons frequently surpasses the six pentagons that are part of the SWCNT-cap, and heptagons occasionally form.

From analysis of the structure during growth, see Supplementary Movies 1, 2, and 3, it was found that, like hexagons, penta- and heptagons form at the tube-catalyst interface. Thus, a distinction is made between interface defects (penta- and heptagons near the tube-catalyst interface) and trapped defects (penta- and heptagons incorporated in the tube wall). Having successfully grown defect-free SWCNTs, see Fig. 2a and Fig. S10, and verified the presence of both penta- and heptagons during the 5th phase of growth, see Fig. 2c, it is concluded that interface defects are effectively healed during the growth process. Figure 2d shows an example of the healing of a pentagon interface defect, while Fig. 2e exemplifies the healing of a more complex pentagon-heptagon pair. From these and Supplementary Movies 2 and 3, key processes involved in the healing of interface defects are identified.

1. Etching of the SWCNT-edge. The removal of carbon atoms from the edge (etching) of the tube is key to exposing interface defects to the catalyst where they can heal.

2. Carbon-carbon bond cleavage. Opening of the ring which forms the interface defect, whether pentagon or heptagon, is essential to heal the defect. This, like etching, requires cleavage of carbon-carbon bonds at the edge of the tube.

3. Stabilization of open rings. There are two ways of healing an interface defect; removing it entirely (etching) or converting it to a hexagon. The latter either requires that open rings are held open long enough so that additional carbon atoms can be added (pentagons → hexagons), or reconfiguration of the edge by the conversion of the heptagon to hexagon as seen in Fig. 2e.

The efficiency of DeepCNT-22 enables growth simulations on time scales much closer to experiment than previously possible, allowing for statistical analysis of defect formation and lifetimes. During growth of the (6,5) SWCNT shown in Fig. 2, a total of 779 unique pentagons were identified, compared to only 28 heptagons. For the pentagons, the time between formation of interface defects, $\delta t$, is plotted in Fig. 2f as a log-log histogram. Here it is clear that $\delta t$ can be modeled using a typical exponential distribution, whose probability density function (PDF) is given by

$$f_{\delta t} = \lambda_1 e^{-\lambda_1 \delta t} \tag{1}$$

The exponential distribution describes the time between events in a Poisson point process, which means that $\delta t$ is stochastic, i.e. the formation of interface defects follows a simple single-barrier process. Fitting the cumulative distribution function (CDF) of Eq. (1) to the normalized cumulative sum of the measured values of $\delta t$ yields $\lambda_1 = 1.08 \times 10^9$ s$^{-1}$. This gives an expected value for the time between formation of interface defects, $\langle \delta t \rangle = \frac{1}{\lambda_1} = 0.925$ ns. From the CDF, it is also evident that there is a 99% probability that interface defects are formed within 4.26 ns of each other.

Like $\delta t$, the lifetime of interface defects, $\tau$, can also be measured. As shown in Fig. 2g, $\tau$ appears linear in the log-log histogram, which is the signature of a power-law distribution, $f_\tau \propto \tau^{-\alpha}$. This distribution is known to be heavy-tailed, meaning that the tail of the power-law distribution is not exponentially bound[42]. However, as seen in Fig. 2g, this is not the case for $\tau$, as there are no interface defects with a lifetime longer than 4 ns. Thus, it is suitable to model $\tau$ as a power-law distribution with an exponential cutoff whose PDF is given by

$$f_\tau = \frac{\lambda_2^{1-\alpha}}{\Gamma(1-\alpha, \lambda_2 \tau_{\min})} \tau^{-\alpha} e^{-\lambda_2 \tau} \tag{2}$$

here $\Gamma(1-\alpha, \lambda_2 \tau_{\min})$ is the upper incomplete gamma function. For details on the derivation of Eq. (2) and its CDF see Section 3 of the Supplementary Information. The power-law distribution describing $\tau$ implies that healing of interface defects is a more complex process than formation, involving multiple steps with individual barriers resulting in stochastic lifetimes. Fitting the CDF of Eq. (2) to the normalized cumulative sum of the measured values of $\tau$ yields $\alpha = 1.20$, $\lambda_2 = 1.04 \times 10^9$ s$^{-1}$ and $\tau_{\min} = 1.10 \times 10^{-12}$ s. This gives an expected value for the lifetime of interface defects, $\langle \tau \rangle = \frac{1}{\lambda_2} \frac{\Gamma(2-\alpha, \lambda_2 \tau_{\min})}{\Gamma(1-\alpha, \lambda_2 \tau_{\min})} = 0.082$ ns and from the CDF, it is found that 99% of all interface defects have a lifetime shorter than 1.17 ns.

To study how $\delta t$ and $\tau$ are influenced by various growth conditions such as growth rate $k$ or temperature $T$, a snapshot was extracted from the growth of the (6,5) SWCNT, Sim. 1 in Table 1, and MD simulations were performed for 1 to 2 μs at different temperatures without adding any carbon atoms to the system, Sim. 2–6 in Table 1. These simulations represent conditions closer to experimental growth, where the growth rate is approximately 50 to 1000 times lower than what was used in Sim. 1. From Table 1, an approximately 7% reduction in $\langle \delta t \rangle$ is observed for the faster-growing SWCNT in Sim. 1 compared to Sim. 4. Longer interface defect lifetimes are also seen for Sim. 1, with an approximately 82% larger $\langle \tau \rangle$ compared to Sim. 4. However, given that the growth rate in Sim. 1 is more than 625 times higher than in Sim. 4, it is concluded that both the time between the formation of interface

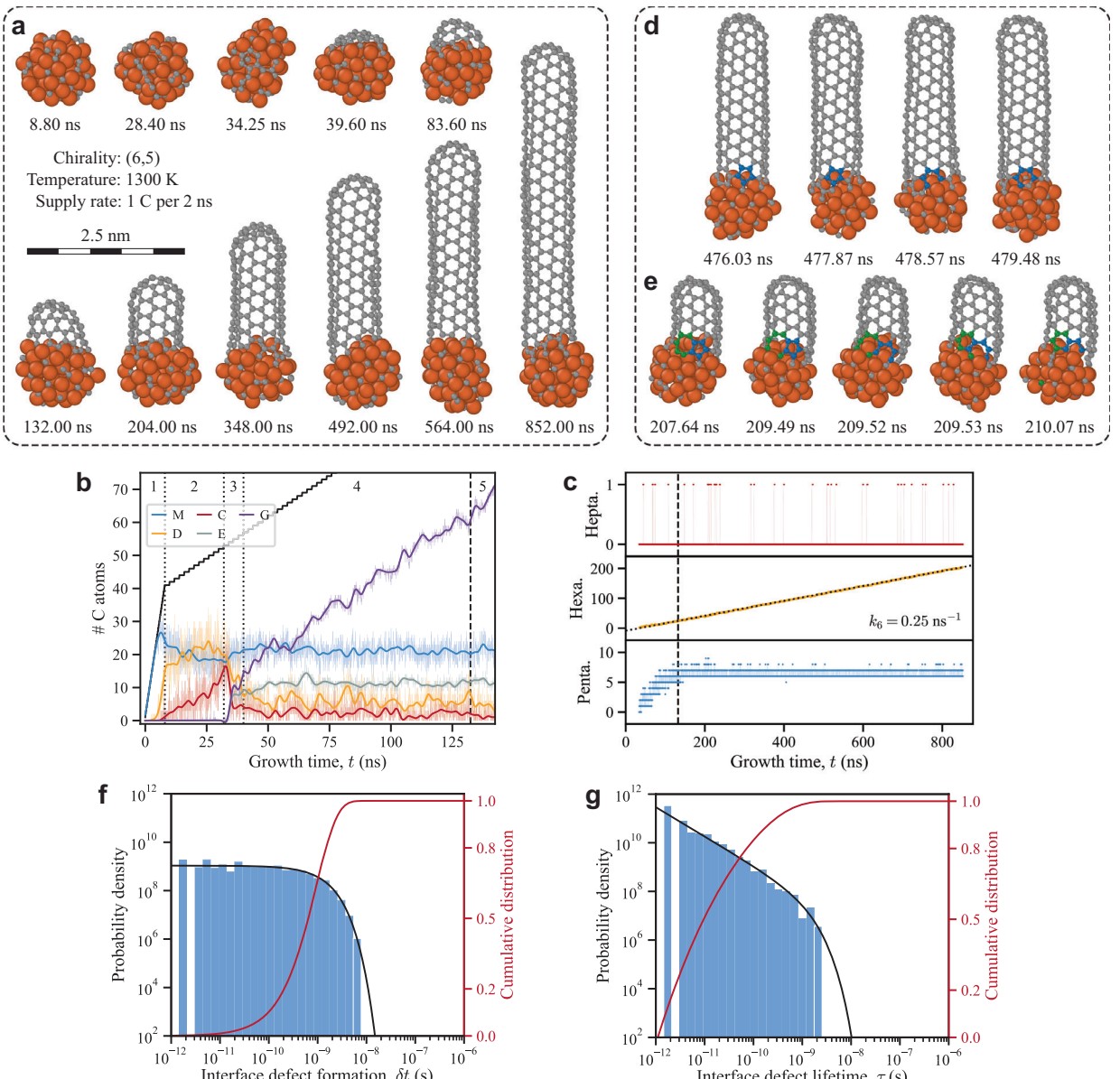

**Fig. 2 | Growth of a defect-free (6,5) single-walled carbon nanotube (SWCNT) on a Fe$_{55}$ catalyst at a temperature of $T = 1300$ K and growth rate of $k = 0.5$ ns$^{-1}$.** Panel (**a**) displays 11 snapshots of the structure during the growth process and panels (**d**, **e**) illustrate the healing of a pentagon and penta-heptagon pair interface defect. The orange and grey spheres represent Fe and C atoms, respectively, in panels (**d**, **e**) blue and green spheres depict C atoms initially belonging to a pentagon and heptagon, respectively. **b** Shows the number of carbon atoms comprising each species, including monomers (M), dimers (D), chains (C), part of the edges (E), and graphitic structures (G), during the early phases of growth. The solid black line is the total number of carbon atoms added to the system, the transparent colored lines represent raw data, the solid lines are the result of applying a low-pass filter and the dotted/dashed vertical lines demarcate each phase of growth labeled by 1, 2, 3, 4 and 5. **c** presents the number of penta-, hexa-, and heptagons during growth, with a linear regression (dotted line) determining the hexagon formation rate, $k_6$. The dashed vertical line in (**b**, **c**) marks the time at which the SWCNT-cap is fully formed, $t = 132.41$ ns. **f**, **g** Show the probability density function (solid black line) and the cumulative distribution function (solid red line) for the time between formation of interface defects, $\delta t$, and the interface defect lifetime, $\tau$, during the growth process after the cap is fully formed. The sample sizes used for the distributions in (**f**, **g**) were 778 formation intervals and 343 lifetimes, respectively. Source data for (**b**, **c**, **f**, **g**) is provided in the Source Data file.

defects and their lifetimes are largely independent of the growth rate. Thus, at these growth rates, the system is close to equilibrium and the etching of the CNT-edge is not significantly affected, enabling defect-free growth.

In contrast, the growth temperature significantly affects both $\delta t$ and $\tau$. By comparing the MD simulations of the extracted snapshot performed at different temperatures, Sim. 2-6 in Table 1, it is evident that as the temperature decreases, $\delta t$ increases significantly. With a 2 to 3 times increase in $\langle \delta t \rangle$ observed for only a 100 K decrease in growth temperature.

Similarly, $\langle \tau \rangle$ increases with a decrease in temperature, although here the effect is less pronounced, with only a 15 to 30% increase for a 100 K decrease in temperature.

## Impact of growth conditions on defect-free growth

For reliable production of long, defect-free CNTs with uniform properties over their entire length, it is crucial to understand how growth rate and temperature affect the entrapment of interface defects. Thus, a qualitative model is proposed for the expected length, in terms of the number of carbon atoms, $\langle N_C \rangle$, that a CNT can reach during growth

**Table. 1 | Interface defect statistics**

| | Growth conditions | | | Interface defect statistics | | | |
|---|---|---|---|---|---|---|---|
| Sim. | $T$ (K) | $k$ (ns$^{-1}$) | $t_{end}$ (ns) | # penta. | # hepta. | $\langle\delta t\rangle$ (ns) | $\langle\tau\rangle$ (ns) |
| 1 | 1300 | 0.5 | 852 | 779 | 27 | 0.925 | 0.082 |
| 2 | 1500 | $< 10^{-3}$ | 1000 | 4649 | 274 | 0.215 | 0.028 |
| 3 | 1400 | $< 10^{-3}$ | 1000 | 2486 | 90 | 0.402 | 0.036 |
| 4 | 1300 | $< 8 \times 10^{-4}$ | 1283 | 1288 | 42 | 0.996 | 0.045 |
| 5 | 1200 | $< 10^{-3}$ | 1000 | 414 | 4 | 2.415 | 0.053 |
| 6 | 1100 | $< 5 \times 10^{-4}$ | 2000 | 238 | 5 | 8.382 | 0.061 |

Data obtained from growth of (6,5) single-walled carbon nanotubes (SWCNTs) on Fe$_{55}$ catalysts at different conditions. Here Sim. represents the different simulations, $T$ the growth temperature, $k$ the carbon supply rate and $t_{end}$ the growth time. # penta., # hepta. are the number of penta- and heptagons formed during growth, respectively. $\langle\delta t\rangle$ and $\langle\tau\rangle$ are the expectation values for the time between interface defect formation and interface defect lifetime, respectively. Note that, Sim. 1 corresponds to the growth of the (6,5) SWCNTs shown in Fig. 2 while Sim. 2-6 correspond to simulations with a constant number of carbon atoms.

before an interface defect is likely to be trapped. As detailed in Section 4 of the Supplementary Information, this model is based on the distributions that model the interface defects, Eqs. (1) and (2), and gives the expected length as

$$\langle N_C \rangle = \frac{k}{\lambda_1} \frac{\Gamma\left(1 - \alpha, \lambda_2 \tau_{min}\right)}{\Gamma\left(1 - \alpha, \lambda_2 \frac{2}{k}\right)} \quad (3)$$

Here, $k$ is the growth rate of the CNT, while $\alpha$, $\lambda_1$, $\lambda_2$, and $\tau_{min}$ are the parameters from Eqs. (1) and (2). Though Eq. (3) accounts for the effect of the growth rate on the expected length, $\langle N_C \rangle$, the impact of growth temperature is absent. This can be addressed by including the temperature effects on $\delta t$ and $\tau$, as demonstrated in Table 1, by modeling the temperature behavior of $\alpha$, $\lambda_1$, $\lambda_2$ and $\tau_{min}$ as shown in Fig. S11. Combined with Eq. (3) it is now possible to construct a qualitative map of defect-free CNT lengths for different combinations of growth rates and temperatures.

The map shown in Fig. 3a reveals two growth regimes with a sharp transition, a light blue region representing growth conditions resulting in defective tubes and a dark blue region representing growth conditions favorable for growing long defect-free tubes. Experimentally, the growth rate of CNTs has been correlated to the partial pressure, $P$, of the carbon feedstock gas (supply of carbon atoms), increasing monotonically with pressure[43–45].

From Fig. 3a it is evident that for a set growth temperature, decreasing the growth rate, i.e., lowering the partial pressure, $P$, results in higher quality CNTs (growth of long defect-free tubes). Likewise, for a set growth rate (partial pressure), increasing the growth temperature will increase the quality of the grown CNTs. These results agree qualitatively with the experimental results of Picher et al.[46] presented in Fig. 3b where the same trends can be found. Independent experimental results from Vinten et al.[47] also directly support this.

Obtaining higher quality CNTs at lower growth rates is easily understood as low growth rates allow more time for defects to heal. However, obtaining higher quality CNTs at higher growth temperatures might seem counterintuitive, given that high growth temperatures decrease $\langle\delta t\rangle$, leading to the formation of more interface defects as shown in Table 1. But the reduction in $\langle\tau\rangle$ at high temperatures decreases the likelihood of these interface defects becoming trapped inside the tube wall during growth, counteracting the increased rate of formation of interface defects. Consequently, if the growth rate (partial pressure) is appropriately chosen to match the growth temperature there is theoretically no upper limit to the length of defect-free CNTs that can be grown. Moreover, higher growth temperatures enable faster growth of long defect-free CNTs, if the carbon supply rate can be controlled. Both can be achieved by carefully tuning the growth

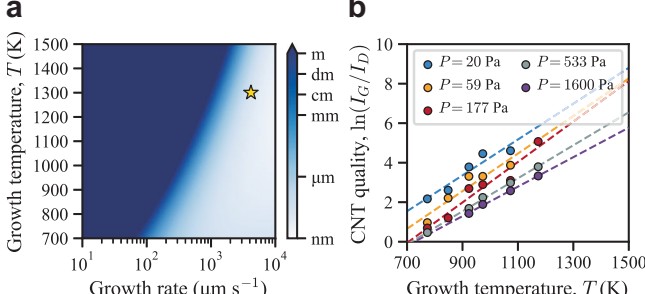

**Fig. 3 | Influence of growth rate and temperature on defect-free carbon nanotube (CNT) growth.** The map in (**a**) shows the expected length that CNTs can grow before an interface defect is trapped. To give a better qualitative understanding of the expected length, the value given by Eq. (3) is converted to meters through multiplication by the length per carbon atom of a (11,3) single-walled CNT ($8.35 \times 10^{-12}$ m per C atom). Here the gold star marks the growth conditions used to grow the (6,5) tube shown in Fig. 2. The plot in (**b**) shows the quality of CNTs grown under different experimental conditions, $T$ and $P$, as determined by the ratio of G-band, $I_G$, and D-band, $I_D$, Raman intensities. Here the markers are reproduced from the published experimental data of Picher et al.[46] and the dashed lines are a linear regression to this data. Source data for this figure is provided in the Source Data file.

conditions to control the decomposition of the precursor gas at the growth temperature while maintaining stable conditions.

## Dynamics of the tube-catalyst interface

As shown, both growth and the formation and healing of interface defects occur at the tube-catalyst interface. Therefore, it becomes crucial to study how the tube-catalyst interface evolves, which has a direct impact on the current understanding of growth mechanisms[48–52]. By tracking the configuration of the SWCNT-edge during growth, the dynamics of the tube-catalyst interface can be studied. For the growth of the (6,5) SWCNT, the complete evolution of the tube-catalyst interface can be observed in Supplementary Movie 4, from which the 9 most common edge configurations are shown in Fig. 4a. These make it evident that the tube-catalyst interface is highly dynamic throughout growth, with a varying number of armchair pairs, $N_A$, and zigzag sites, $N_Z$, and does not evolve in a continuous spiral growth mode[48].

To compare the evolution of the tube-catalyst interface, an edge chiral index $(n_e, m_e)$ is derived, where $n_e = N_A + N_Z$ and $m_e = N_A$. Identifying edges with the same number of armchair pairs and zigzag sites as a perpendicularly cut tube with chirality $(n,m)$ becomes straightforward with this approach, as $n_e = n$ and $m_e = m$ in these instances. Figure 4b shows the distribution of the edge chiral index, after the formation of the SWCNT-cap, for all defect-free SWCNTs grown in this work.

Intriguingly, the most dominant edge chiral index does not necessarily match the chirality of the grown tube. For the (6,5) SWCNT, the most dominant edge, with a probability of 43.3%, is $(n_e, m_e) = (8,3)$, closely followed by $(n_e, m_e) = (7,4)$ with a probability of 37.3%. These two edge chiral indices account for 80.6% of all edge chiral indices observed during growth, hinting at the importance of the configurational entropy of the SWCNT-edge[50]. This drives the edge to be chiral, regardless of the tube chirality, as can be seen by comparing the $(n_e, m_e)$ distributions for the (7,7) and (9,5) SWCNTs in Fig. 4b.

The edge chiral index, however, does not uniquely identify a SWCNT-edge, as there are multiple ways to arrange $N_A$ and $N_Z$. Confirming the importance of configurational entropy thus requires checking whether a preferred edge configuration or set of configurations emerges during growth. This is done by counting the occurrence of each unique edge configuration, accounting for the cyclic nature of

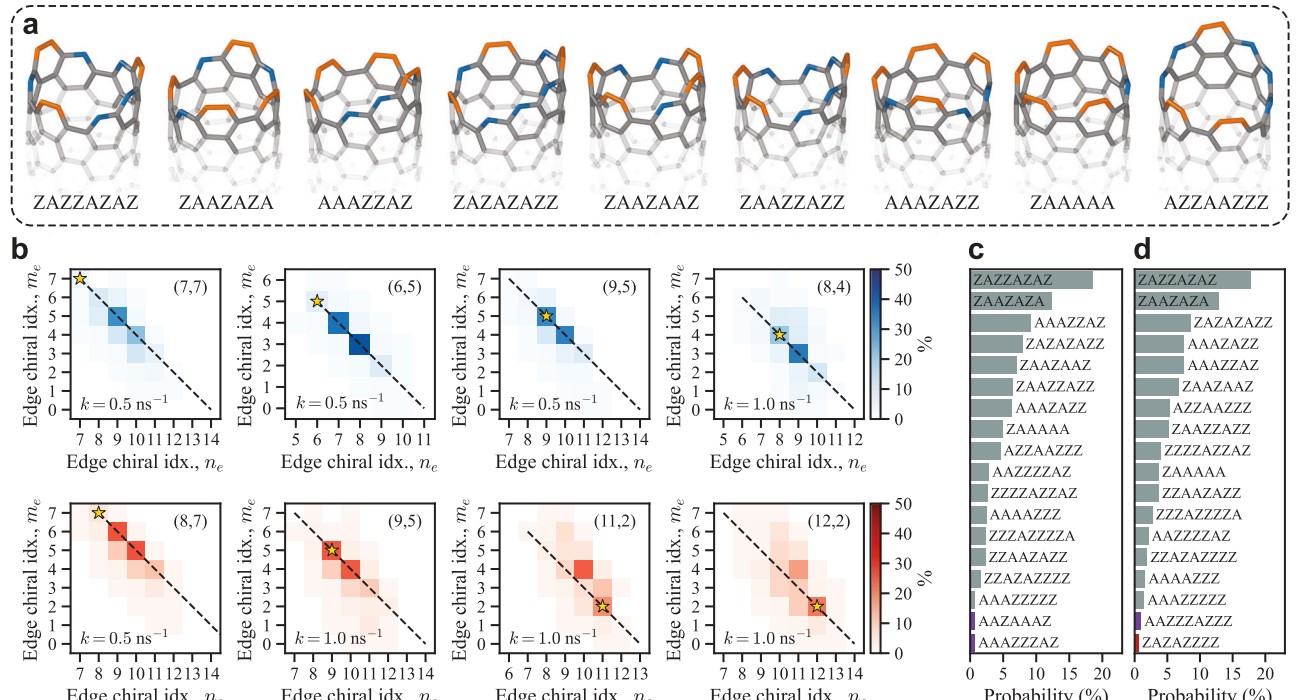

**Fig. 4 | Edge configurations observed during growth of single-walled carbon nanotubes (SWCNTs) on Fe₅₅ catalysts.** Panel (**a**) shows the 9 most common edge configurations observed during growth of the (6,5) SWCNT in Fig. 2. Here zigzag sites are denoted by Z and colored blue, while armchair pairs are denoted by A and colored orange. **b** 2D histograms showing the distribution of edge chiral indices $(n_e, m_e)$ for different SWCNTs grown at 1300 K (blue) and 1500 K (red) and different growth rates $k$ as marked in the histograms. Here the chirality $(n,m)$ of the grown tube is shown in the upper right corner of each histogram and marked by the gold star. Dashed lines show where the length of the edge, $n_e + m_e$, matches the length of an edge of a perpendicularly cut tube with the same chirality. The sample size used for each histogram was as follows; (7,7): 149,360, (6,5): 287,129, (9,5): 171,279, (8,4): 72,222, (8,7): 155,575, (9,5): 74,247, (11,2): 58,180 and (12,2): 54,885 edge chiral indices. The 18 most observed edge configurations during (**c**) the entire growth simulation of the (6,5) SWCNT and (**d**) just before the formation of an interface defect. Here the color represents the length of the edge where red: 10, grey: 11, purple: 12 atoms. The sample size used for (**c**, **d**) was 287,129 and 620 edge configurations, respectively. Source data for (**b**–**d**) is provided in the Source Data file.

the edge. As shown in Fig. 4c, the most frequently observed edge configuration during growth of the (6,5) SWCNT is ZAZZAZAZ with a probability of 18.5%, closely followed by ZAAZAZA (12.4%), AAAZZAZ (9.18%), and so on. Thus, there is no preferred edge configuration or set of configurations during growth, confirming the importance of configurational entropy—which has not only been shown to affect stability[50] but also indirectly evidenced via dynamic instabilities in experimentally measured growth kinetics[52,53]. Additional data on the most frequently observed edge configurations for the other defect-free SWCNTs grown can be found in Fig. S12.

By comparing the edge configurations present just before the formation of interface defects, Fig. 4d, to those of all the edges seen during growth, Fig. 4c, it is apparent that formation of interface defects does not depend on the configuration of the edge but is instead purely stochastic. Similarly, there is no apparent correlation between the configuration of interface defects, at the time of formation, and their lifetimes as shown in Fig. S13. Thus, the configuration of the interface defect, at the time of formation, does not determine how it heals, resulting in stochastic lifetimes.

## Discussion
The quality of the DeepCNT-22 MLFF and its ability to drive long-timescale simulations, enabled us to probe the dynamics of growing carbon nanotube interfaces. Large fluctuations in armchair and zigzag edge atoms were observed during growth which demonstrates the importance of configurational entropy, affecting both their ordering and numbers. The formation and healing of defects are shown to depend on the interplay between the growth rate and temperature, paving the way for the controlled growth of long,

defect-free CNTs. Achieving such growth, with precise control of growth temperatures and rates, may justify moving away from traditional hot-wall CVD synthesis and innovative new methods of supplying carbon to the catalysts, a direction for future experiments. On the theory side, this renewed understanding of growth mechanisms should in the future be extended to elemental or alloyed catalysts that remain stiffer and less compliant during growth, which may promote chiral selectivity[54].

## Methods
To create the DeepCNT-22 dataset, an initial set of structures was generated using various methods, including molecular dynamics (MD) driven by density functional tight binding, randomly perturbed structures, and carbon allotropes from the GAP-20 dataset[55]. After which the dataset was further refined using a variant of the active learning scheme[56–58], in which an ensemble of machine learning force fields (MLFFs) is trained on the dataset and employed to drive MD simulations of single-walled carbon nanotube (SWCNT) growth. During this process, the deviation in the MLFFs' force predictions (i.e., model deviation) is utilized to identify unrepresented structures that emerge during the growth process, which are then labeled and added to the dataset such that a new ensemble of MLFFs can be trained. This procedure is repeated until the model deviation remains low throughout the growth simulation. Regardless of the generation method, all structures were labeled with energies and forces obtained via dispersion-corrected density functional theory (DFT) calculations.

After training, the DeepCNT-22 MLFF was used to drive MD simulations of SWCNT growth. Supplementary Movies 1–4 were then

generated from the MD trajectory of the grown (6,5) SWCNT. Post-growth, the SWCNT structure was adjusted to align its axis parallel to the z-axis. Movies were subsequently rendered from the aligned MD trajectory using the OVITO software package[59]. The Smooth trajectory modifier, available in OVITO, with a window size of 5 was applied to minimize thermal vibrations and highlight the evolution of the structure during growth. Visualization of the tube-catalyst interface as shown in Supplementary Movie 4, involved removing all iron atoms and then iteratively removing carbon atoms with a coordination number less than 2 until only those with a coordination number of 2 or higher remained.

## Density functional tight binding

The initial dataset for DeepCNT-22 includes structures obtained from density functional tight binding (DFTB) MD simulations of SWCNT nucleation originating from atomic carbon precursors on Fe nanoparticle catalysts. DFTB is an extended two-center Hückel approximation to DFT, employing a minimal Slater-type all valence basis set. This allows dynamic simulations to occur orders of magnitude faster than DFT, while including electronic effects not found in classical force field-based methods. MD simulations relied on self-consistent charge DFTB (SCC-DFTB)[60] to compute quantum chemical potential energy and energy gradients during each MD iteration. The trans3d-0-1 parameter set was used[61], with all simulations conducted within the DFTB+ software package[62] version 21.1. Newton's equations of motion were integrated using the velocity-Verlet algorithm[63], with a 1.0 fs time step and a finite electronic temperature of 10,000 K[64–66]. A canonical NVT ensemble was maintained at 1500 K using a Nosé-Hoover chain thermostat[67–69] of length 3.

Structures that were procured from MD simulations include $Fe_{13}$, $Fe_{38}$, or $Fe_{55}$ nanoparticles within a periodic cell without C atoms, or with 20, 30, or 40 C atoms for the case of $Fe_{13}$. Extracted structures from these simulations featured Fe nanoparticles with surface-adsorbed carbon monomers and dimers, carbon chains and junctions, ring networks frequently containing defects, and SWCNT-cap and tube-like structures, consistent with previous DFTB growth simulations[70,71]. DFTB MD simulations were also used to anneal high-energy structures obtained by early versions of the MLFF, with the resulting structures added to the dataset. To identify which structures from the DFTB MD simulations to label with DFT and include in the training data, farthest point sampling was conducted on the DFTB calculated potential energies.

## Density functional theory

DFT calculations were performed using the Vienna Ab initio Simulation Package (VASP)[72–74] version 6.3.0. A plane wave basis set was employed, and the projector-augmented wave method[75,76] was utilized with standard pseudopotentials (Fe 06Sep2000 and C 08Apr2002). The optB86b-vdW van der Waals density functional[77,78] was selected to account for dispersion interactions. High precision (PREC = Accurate) was employed throughout the calculations, with a plane wave cutoff energy of 600 eV (ENCUT = 600) and no symmetry constraints applied (ISYM = 0). To ensure accuracy, the electronic self-consistent loop converged to a tolerance of $10^{-6}$ eV (EDIFF = 1.0E-6). Gaussian smearing (ISMEAR = 0) was utilized with a smearing width of 0.05 eV (SIGMA = 0.05) to assist in the convergence of the calculations. Spin-polarized calculations were conducted (ISPIN = 2), with a high initial magnetic moment, $3 \mu_B$, assigned to each Fe atom. For all periodic structures, a Γ-centered k-point mesh with a density of 0.25 Å$^{-1}$ (KSPACING = 0.25) was used, while for non-periodic structures, only the Γ-point was used with a minimum of 10 Å vacuum spacing between periodic images. Only single point calculations were performed, as DFT calculations were utilized to label the training data.

## Machine learning force field

DeepCNT-22 is built on the Deep Potential-Smooth Edition architecture[79] and was developed using DeePMD-kit[29] version 2.1.1. This MLFF is of the Behler-Parrinello type[80], wherein the energy of each atom in a structure is predicted using a neural network, and subsequently summed to yield the total energy of the structure. A type map of [Fe, C] was utilized together with the type embedding approach, which improves performance and accuracy by allowing the use of a single descriptor embedding net and fitting net shared by both atom types. For further information on the Deep Potential-Smooth Edition architecture and the type embedding approach, consult the DeePMD-kit documentation[81].

Utilizing the type embedding approach, an embedding net with 2 hidden layers containing 8 neurons each was employed. The descriptor embedding net was of type se_e2_a and consisted of 3 hidden layers with 16, 32, and 64 neurons, as well as 8 axis neurons. A cutoff of 5.0 Å was applied to define each atom's local environment, with a smooth cutoff of 0.5 Å, and a fitting net comprising 3 hidden layers with 256 neurons each was used. The GELU activation function[82] was applied for each hidden layer, and no timestep was used in the ResNet architecture[83]. During training, the following loss function was applied, $\mathscr{L} = \frac{p_\epsilon}{N}\Delta E^2 + \frac{p_f}{3N}|\Delta \boldsymbol{F}|^2$, where $N$ denotes the number of atoms, $E$ the energy, and $\boldsymbol{F}$ the forces acting on each structure. Energy and force error weights, $p_\epsilon$ and $p_f$, were set to 0.1 and 1.0, respectively, and remained constant during training. Training was performed for 300,000 batches, using a batch size of 5 structures and the Adam optimizer[84] with an initial learning rate of $10^{-3}$, which decayed exponentially to $10^{-5}$ by the end of the training.

## Molecular dynamics

MD simulations of SWCNT growth were performed using the Large-scale Atomic/Molecular Massively Parallel Simulator (LAMMPS)[85] version 29 Sep 2021 – Update 3, with the deepmd pair style and the DeepCNT-22 MLFF. The nsq algorithm was employed for neighbor list construction with a cutoff distance of 5.0 Å, as it offers slight performance advantages for smaller systems. A 2.0 Å skin distance was incorporated, and the neighbor list was only rebuilt if at least one atom moved more than half the skin distance. Simulations took place in the NVT ensemble using a Nosé-Hoover chain thermostat[67–69] of length 3, and a temperature damping parameter of 0.1 ps. The equations of motion were integrated with a 2.0 fs timestep, maximizing performance while maintaining simulation stability. Initial velocities were drawn from a Gaussian distribution, and the resulting ensemble of velocities had linear and angular momenta zeroed before being scaled to correspond to the growth temperature $T$. Fe and C atom masses were set to 55.847 u and 12.011 u, respectively. C atoms were introduced individually at a rate of $k$ ns$^{-1}$ within a spherical deposit region of radius $\frac{d_C}{4}$ located at the center of the simulation box, here $d_C$ is diameter of the Fe catalyst. To guarantee carbon atoms were consistently deposited inside the Fe catalyst, the system was recentered after every timestep, ensuring that the catalyst remained at the center of the simulation box. The number of degrees of freedom contributing to the system temperature was dynamically updated to account for the newly deposited carbon atoms. Simulation data, including the number of 0, 1, 2, and 3 carbon-carbon coordinated atoms, total number of carbon atoms added, and atomic coordinates, and carbon-carbon coordination numbers, was recorded to file every 2 ps for subsequent analysis.

## Reporting summary

Further information on research design is available in the Nature Portfolio Reporting Summary linked to this article.

## Data availability

The DeepCNT-22 MLFF, dataset, and the full trajectory from the growth of the (6,5) SWCNT generated in this study along with the

inputs used to label data with VASP, train DeepCNT-22 with DeePMD, and run MD simulations with LAMMPS have has been deposited in a Zenodo repository under https://doi.org/10.5281/zenodo.10215578 [https://zenodo.org/records/10215578]. The DFT relaxed structures generated in this study are provided as a Supplementary Data 1. Source data are provided with this paper.

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

## Acknowledgements

The authors would like to acknowledge the computational resources provided by the Institute for Basic Science (Korea) at the compute clusters Cimulator (CMCM, Ulsan) and Olaf (IBS-HQ, Daejeon). As well as the computational resources provided by the Swedish National Infrastructure for Computing via the SNIC 2022/3-29 and SNIC 2022/5-110 projects, partially funded by the Swedish Research Council through grant agreement no. 2018-05973. D.H., B.M. and F.D.

acknowledge financial support from the Institute for Basic Science Korea (IBS-R019-D1). CB acknowledges financial support from the French Agence Nationale de la Recherche (ANR-20-CE09-0007-01). SM acknowledges financial support from the Japan Society for the Promotion of Science KAKENHI (JP23H00163, JP23H00174, JP23H05443) and from the Japan Science and Technology Agency CREST (JPMJCR20B5). J.A.L. acknowledges the Knut and Alice Wallenberg Foundation, and Kempestiftelserna for their financial support. We especially thank Prof. Vincent Jourdain for providing the data from ref. 46 used to produce Fig. 3b.

## Author contributions

Investigation: D.H., B.M.; Methodology: D.H., B.M.; Data curation: D.H., B.M.; Formal analysis: D.H.; Validation: D.H.; Conceptualization: D.H., B.M., C.B., S.M., J.A.L. and F.D.; Project administration: D.H., F.D.; Funding acquisition: F.D.; Resources: F.D., J.A.L.; Supervision: F.D.; Visualization: D.H.; Writing - original draft: D.H., BM; Writing - review & editing: D.H., B.M., C.B., S.M., J.A.L. and F.D.

## Funding

## Competing interests

The authors declare no competing interests.
