## [Peer Review File · Nature Communications]

Dynamics of growing carbon nanotube interfaces probed by machine learning-enabled molecular simulationsREVIEWER COMMENTS

Reviewer #1 (Remarks to the Author):

The authors developed a ML potential to enable long-time scale MD simulations of SWNT growth, with an accuracy approaching DFT accuracy. This effort is in line with current efforts in the MD community.

In spite of decades of research on CNT growth mechanisms, including extensive computational work, there are still plenty of open questions. In this work, the authors address defect-free growth of long tubes on "long" time scales. They succeeded in their efforts to a remarkable extent.

The work is definitely novel, it is very appealing and of interest to the community, and all conclusions are supported by the data provided. Within the boundary conditions of the simulations, I have no reason to doubt the results.

However, I would like to ask the authors to consider the following two remarks / questions, and possibly modify the manuscript accordingly.

1) to the best of my understanding, the tubes in the simulations (at least, those that are not subject to one of the various failure mechanisms such as cap liftoff failure etc) grow so long because of the effective and efficient defect healing process. However, to what extent could the defect-free growth also be a consequence of having an insufficient number of "amorphous" structures in the ML training set? That is, if the training set only considers "good" structures, that lead to perfect growth, that this is also what the simulation will eventually result in. Did the authors also include numerous highly defective structures in the training set?

2) if I am not mistaken, the growth is initiated by allowing C-atoms to be incorporated in the structure. In reality, it would rather be hydrocarbon molecules dissociating on the surface of the catalyst. The hydrogen present might affect the growth process. This is not accounted for. While this is the case in many MD CNT growth simulations (although there are simulation studies in the literature on simulated CNT growth from hydrocarbons), could the authors comment on the effect of this effect not being taken into account?

Finally, I would also like to remark that I do not fully agree that "DeepCNT-22 enables growth simulations on experimentally relevant time scales" (line 142 in the manuscript) - the growth rate is still $\sim 1000x$ faster than in reality. How does that affect the stochastic nature of the process? I.e., I would expect entropy (i.e., the system not having sufficient time to find the optimal configuration) to lead to defective structures in such case (without biasing) - unless the training set does not allow this: see question 1 above. Could the authors additionally answer this?

Reviewer #2 (Remarks to the Author):

In this manuscript, the authors leverage the machine learning force fields to develop DeepCNT-22 and explore the nucleation and growth mechanism of carbon nanotubes (CNTs) with much higher efficiency and accuracy than conventional MD. Using DeepCNT-22, the authors unveil the highly dynamic characteristics of tube-catalyst interface and demonstrated the significance of the configuration entropy. It is pointed out that defects are formed at the tube-catalyst interface in a stochastic manner. The authors also demonstrated that defect-free CNTs growth can be achieved by engineering the formation and lifetime of CNTs. Under low growth rates and high temperatures, the healing outweighs formation and defect-free growth can be achieved. This DeepCNT-22 model will have significant impact on the CNT community for designing the path towards the chirality-controlled growth of SWCNTs. However, there are still some major and minor issue that needs to be addressed.

1. In the real growth condition, hydrogen plays an important role to the nucleation and growth of CNTs. Though the DeepCNT-22 model was considering the steps after C is already dissolved in Fe, the surrounding hydrogen radicals may still affect the CNT-Fe interface. The authors should evaluate the role of hydrogen from either the decomposition of hydrocarbon supply or hydrogen gas.

2. The formation interval δt and lifetime τ of defect both span several orders in the time scale. By analyzing the probability density function, the authors think the defect formation and healing are both stochastic process. However, there is lack of the understanding of a physical picture behind. It is also possible that there is a distribution of defect-catalyst-interface configuration that present different time scales of formation and healing. Is it possible to classify the defect-catalyst interface. The authors should provide more evidence to evaluate the defect formation and healing.

3. According to the manuscript, the defect lifetime is largely independent of the growth rate or carbon supply. But if we consider a faster growth rate (like 5 times higher growth rate), it means the driving force by the Carbon-atom chemical potential between the Fe-C and CNT lattice is higher (5 times higher). In intuition, the large driving force should significantly suppress the etching of CNT. In this view, the defect lifetime should be largely affected by the growth rate.

4. Fig. 3a quite deviates from the experiments. The fastest reported growth rate is about 100 $\mu\text{m/s}$. According to Fig. 3a, all the growth of CNT should be defect free. The similar with the last question, is it possible the defect lifetime is undervalued?

5. The author should provide direct simulation result to validate the $\langle N_c \rangle$. The authors could further increase the carbon supply to obtain a shorter $\langle N_c \rangle$ to directly simulate the defect trapping and provide the relation between $\langle N_c \rangle$ and temperature, growth rate.

minor comments:

6. In the Abstract, "Here, using molecular dynamics simulations driven by a machine learning force field¹³ (MLFF) we developed, DeepCNT-22, " seems not proper. It sounds like the authors developed the MLFE all themselves.

7. In the Abstract, "This contradicts the previous notion of a continuous spiral growth mode¹⁴, but confirms that the growing tube edge exhibits significant configurational entropy¹⁵" is not accurate. The continuous spiral growth mode depends on the practical growth condition. With moderate etching species, the growth is chiral-angle dominant. (ACS Nano 2022, 16, 5627–5635)

8. In the Abstract, "healing becomes more efficient than formation, allowing CNTs to grow defect-free to seemingly unlimited lengths" is not accurate. Whether defect trapping or defect free growth should be attribute to the competing of defect healing rate (lifetime) and the lattice growth rate. This sentence is very misleading.

9. Page 7, "From Fig. 2b, it ...", the later claims cannot be directly seen from Fig. 2b.

10. There are quite a few grammatical errors and should be corrected. For example, in line 56, the '. Which' should be changed into ', which'. In line 71, 'understand for' should be changed into 'understand'. In line 197, 'The map in shown in' should be changed into 'The map shown in'. In Fig. 2 caption, the label C should be lower case.

Reviewer #3 (Remarks to the Author):

We sincerely appreciate the insightful feedback from the reviewers, which has substantially improved our manuscript. We have highlighted all changes in the manuscript text file using color highlighting for ease of reference. Overall, we believe that our responses and revisions comprehensively address the concerns raised by the reviewers.

Below are our point-by-point responses to each of the reviewers' comments structured as follows.

Remark X.Y

Comment of the reviewer.

Response X.Y

Response of the authors.

Action taken when applicable.

Reviewers' comments:

Reviewer #1 (Remarks to the Author):

The authors developed a ML potential to enable long-time scale MD simulations of SWNT growth, with an accuracy approaching DFT accuracy. This effort is in line with current efforts in the MD community.

In spite of decades of research on CNT growth mechanisms, including extensive computational work, there are still plenty of open questions. In this work, the authors address defect-free growth of long tubes on "long" time scales. They succeeded in their efforts to a remarkable extent.

The work is definitely novel, it is very appealing and of interest to the community, and all conclusions are supported by the data provided. Within the boundary conditions of the simulations, I have no reason to doubt the results.

However, I would like to ask the authors to consider the following two remarks / questions, and possibly modify the manuscript accordingly.

Remark 1.1

-To the best of my understanding, the tubes in the simulations (at least, those that are not subject to one of the various failure mechanisms such as cap liftoff failure etc) grow so long because of the effective and efficient defect healing process. However, to what extent could the defect-free growth also be a consequence of having an insufficient number of "amorphous" structures in the ML training set? That is, if the training set only considers "good" structures, that lead to perfect growth, that this is also what the simulation will eventually result in. Did the authors also include numerous highly defective structures in the training set?

Response 1.1

We thank the reviewer for the comments on our manuscript. While it is difficult to comprehensively show all the structures included in the training data, we do indeed include a large amount of highly defective structures. This is necessary in order for the machine learning force field (MLFF) to learn that these "amorphous" structures have high energy and are thus unlikely to appear during growth. Fig. R1a shows a few representations of defective structures that are in the training data which include very high energy "clustered" carbons, "triangle" defects and defective tubes.

Interestingly these highly defective structures (particularly the "clustered" carbons and "triangle" defects) appear during the early iterations of the active learning (AL) processes. After these are added to the training data, the subsequent iterations of AL show significantly improved quality of the graphitic structure formed on the catalyst, as shown in Fig. R1b going from gen. 3 to gen. 4 of the MLFF. It is important to note however that defective structures (pentagons, heptagons) continue to appear during the AL process due to the high carbon supply rate used ($k = 1, 2, 4$ or 8 C per ns). Its only when a lower carbon supply rate is used during the subsequent production simulations that defect-free growth is achieved.

Figure R1. **a** Examples of different types of defective structures in training data. **b** Example of the MLFF learning the energy of highly defective structures during active learning.

We argue that including such defective structures in the training data improves the generalizability of DeepCNT-22 and prevents biasing the MLFF towards defect-free growth. To verify this, we checked the model deviation for the time steps where interface defects are present during the growth of the (6,5) SWCNT shown in the main text. These are then compared with the model deviation at the time steps where no interface defects occur. As seen in Fig. R2, the model deviation for both is very similar, indicating that DeepCNT-22 has learned to represent tube-catalyst interfaces with and without interface defects to similar levels of accuracy.

Thus, we have a high level of confidence that DeepCNT-22 does not bias toward defect-free growth and that the perfect tubes shown in this work are a result of efficient interface defect healing and not bias in the MLFF. We would also like to point out that, as the reviewer mentioned above, we do not always grow defect-free tubes. As shown in Fig. S6, 174 of the 280 growth simulations performed to study the chirality distribution resulted in failed/defective tubes.

Figure R2. Model deviation during growth of the (6,5) tube for time steps with interface defects (orange) and without interface defects (blue).

To convey to the reader that DeepCNT-22 is accurate throughout growth and does not bias towards defect-free growth, we have added Fig. R2 to the Supplementary information as Fig. S1c along with an accompanying paragraph on page 2.

"In addition to evaluating DeepCNT-22 on a test data set, the accuracy of the MLFF was continuously evaluated during each growth simulation using the model deviation, ϵ [4]. As shown in Fig. S 1c the distribution of ϵ for

timesteps with interface defects (orange) and those without (blue) show only a single peak centered around 250 meV/Å respectively. DeepCNT-22 thus maintains accuracy throughout the growth simulations and shows no bias towards defective or defect-free growth.”

We have also added the following sentence to the main text on page 7 to highlight this.

“In addition, the accuracy of the MLFF was continuously monitored during the MD simulations via model deviation³⁴, which, as seen in Fig. S1c, has a single peak centered around 250 meV/Å. Thus, accuracy is maintained throughout the growth simulation with no bias.”

Remark 1.2

-If I am not mistaken, the growth is initiated by allowing C-atoms to be incorporated in the structure. In reality, it would rather be hydrocarbon molecules dissociating on the surface of the catalyst. The hydrogen present might affect the growth process. This is not accounted for. While this is the case in many MD CNT growth simulations (although there are simulation studies in the literature on simulated CNT growth from hydrocarbons), could the authors comment on the effect of this effect not being taken into account?

Response 1.2

The reviewer is correct that often hydrocarbon precursors are the primary initial source of carbon, particularly during catalytic chemical vapor deposition. Prior molecular dynamics (MD) simulations have demonstrated the dissociation of hydrocarbon species on catalyst surfaces such as Fe into the active growth species of carbon monomers and carbon dimers, as we discuss below. Ultimately, whether carbon comes from one of many commonly used hydrocarbon precursors, it will dissolve into the nanoparticle (bulk or subsurface) as C-H bonds cleave and hydrogen desorbs from the surface. Upon saturation, carbon will diffuse from within the catalyst to the surface, where nucleation will occur as observed here. We do make an addition to the manuscript to mention this important point.

Simulations of CH_x decomposition on a Fe nanoparticle demonstrate that C-H bond scission occurs on the catalyst surface to form Fe-H bonds. Hydrogen atoms subsequently desorb from the surface into the gas-phase in favor of saturating the catalyst with carbon (Eveleens, *Nanoscale*, 2017, 9, 1727-1737). Indeed, these simulations, as well as others for Fe catalysts (McLean, *J. Appl. Phys.*, 2021, 129, 044302; Wang, *Carbon*, 2014, 72, 22-37; Eveleens, *Carbon*, 2019, 146, 535-541) have shown that prior to desorption, hydrogen passivates dangling carbon bonds. This slows C-C bond formation and ring condensation, promoting the formation of hexagons over time as opposed to defective rings like pentagons and heptagons. However, the active carbon species for ring nucleation and subsequent growth of graphitic networks remains as carbon, and it has been demonstrated with DFT calculations that the carbon monomer is the most active species for surfaces like Fe with high carbon affinity (Wang, *Nanoscale*, 2017, 9, 11584-11589; Shu, *Nanoscale*, 2015, 7, 1627).

Finally, recent MD simulations of CH₄ decomposition on Fe₅₅ nanoparticles (Lei, *J. Phys. Chem. Lett.*, 2023, 14, 4266-4272) demonstrate that the decomposition of CH₄ to C is characterized by small energy barriers, and while a high hydrogen surface coverage delays CH₄ dehydrogenation, hydrogen can desorb as H₂, as previously observed. The early stages of the growth process we report here i.e., growth phases 1 and 2, are also observed, whereby monomers present on the surface convert into dimers and carbon chains.

We have added the following paragraph discussing the effect of hydrogen to the main text on page 8.

“Note that here, carbon atoms are supplied directly inside the catalyst, which then diffuse rapidly to the surface, rather than via hydrocarbon (CH_x) decomposition. Previous MD simulations³⁹ have shown that CH_x undergoes C-H bond cleavage on the catalyst to form Fe-H bonds which effectively deposits C on the surface, a process with low energy barriers⁴⁰. While the presence of hydrogen may passivate dangling carbon bonds and slow nucleation in the early stages of growth, hydrogen ultimately desorbs from the surface in favor of saturating the catalyst with carbon.”

Remark 1.3

-Finally, I would also like to remark that I do not fully agree that “DeepCNT-22 enables growth simulations on experimentally relevant time scales” (line 142 in the manuscript) – the growth rate is still ~1000x faster than in reality. How does that affect the stochastic nature of the process? I.e., I would expect entropy (i.e., the system not having sufficient time to find the optimal configuration) to lead to defective structures in such case (without biasing) – unless the training set does not allow this: see question 1 above. Could the authors additionally answer this?

Response 1.3

We agree with the reviewer that the sentence “DeepCNT-22 enables growth simulations on experimentally relevant time scales” may be misleading to the reader and have thus changed it, as well as a sentence on line 74 that used similar language. The question of how the relatively high growth rate used in our simulations (~1000x faster than experiment) affects the stochastic nature of the formation and healing of defects is interesting and something we have considered.

As mentioned in the manuscript, we extracted a snapshot from the growth simulation of the (6,5) tube and ran long timescale MD simulations without adding or removing C atoms from the system. From these simulations we extracted the interface defect formation interval, δt , and lifetime, τ . As shown in Extended Data Table 1, the time between formation of interface defects at $k = 0.5 \text{ ns}^{-1}$ equals $\langle \delta t \rangle = 0.925 \text{ ns}$, which is close to that at the low ($k < 8 \cdot 10^{-4} \text{ ns}^{-1}$) growth rate, $\langle \delta t \rangle = 0.996 \text{ ns}$. Moreover, the growth rate only marginally affects the interface defect lifetimes, τ , where the $k = 0.5 \text{ ns}^{-1}$ growth rate resulted in $\langle \tau \rangle = 0.082 \text{ ns}$ and the low ($k < 8 \cdot 10^{-4} \text{ ns}^{-1}$) growth rate yielded $\langle \tau \rangle = 0.045 \text{ ns}$. While $\langle \tau \rangle$ at $k = 0.5 \text{ ns}^{-1}$ is almost double that at the low growth rate, we must account for the fact that the difference in growth rate is very large, $\frac{0.5}{<8 \cdot 10^{-4}} > 625$. In other words, a more than 625-fold increase in the growth rate only increases the interface defect lifetimes by about 82%. This is a small change given that 0 C atoms were added to the system for the simulation at $k < 8 \cdot 10^{-4} \text{ ns}^{-1}$ compared to 426 C atoms added for the simulation at $k = 0.5 \text{ ns}^{-1}$.

Thus, at the carbon supply rate used in our simulations we are close enough to equilibrium that the etching of the CNT-edge is not significantly affected. Thus, we conclude that at the growth rates considered here ($k \leq 0.5 \text{ ns}^{-1}$) there is no meaningful impact on the stochastic nature of the formation and healing of interface defects.

We have changed the sentence on line 163 to read “The efficiency of DeepCNT-22 enables growth simulations on time scales much closer to experiment than previously possible, allowing a statistical analysis of defect formation and lifetimes.”

We have changed the sentence on line 78 to read “An emerging and powerful method for modeling materials at length and timescales that approach experiment is machine learning force fields²⁶ (MLFFs).”

Reviewer #2 (Remarks to the Author):

In this manuscript, the authors leverage the machine learning force fields to develop DeepCNT-22 and explore the nucleation and growth mechanism of carbon nanotubes (CNTs) with much higher efficiency and accuracy than conventional MD. Using DeepCNT-22, the authors unveil the highly dynamic characteristics of tube-catalyst interface and demonstrated the significance of the configuration entropy. It is pointed out that defects are formed at the tube-catalyst interface in a stochastic manner. The authors also demonstrated that defect-free CNTs growth can be achieved by engineering the formation and lifetime of CNTs. Under low growth rates and high temperatures, the healing outweighs formation and defect-free growth can be achieved. This DeepCNT-22 model will have significant impact on the CNT community for designing the path towards the chirality-controlled growth of SWCNTs. However, there are still some major and minor issue that needs to be addressed.

Remark 2.1

-In the real growth condition, hydrogen plays an important role to the nucleation and growth of CNTs. Though the DeepCNT-22 model was considering the steps after C is already dissolved in Fe, the surrounding hydrogen radicals

may still affect the CNT-Fe interface. The authors should evaluate the role of hydrogen from either the decomposition of hydrocarbon supply or hydrogen gas.

Response 2.1

We thank the reviewer for the comments on our manuscript and agree that hydrogen plays an important role during CNT growth, see our response to Reviewer 1's comment (Response 1.2) regarding the role of hydrogen from precursor decomposition; that the dehydrogenation of hydrocarbon precursors on Fe has low energy barriers and readily occurs. We make an addition to the manuscript to point out this important fact. If the hydrogen concentration is high enough, re-hydrogenation mechanisms become active. These mechanisms involve surface-bound hydrogen from either hydrocarbon decomposition or adsorbed hydrogen radicals from the gas phase. The hydrogen reacts with adsorbed carbon species such as carbon chains and emerging pentagon-hexagon networks. As a result, H-passivated structures form on the surface. The precursor supply rate would need to be very fast, or the growth temperature low, to see such structures favored.

Indeed, while CNTs ultimately nucleate from carbon active species such as monomers and dimers, the role of hydrogen is not trivial and influences the growth rate of graphitic species on the catalyst surface. We aim to incorporate hydrogen into a future MLFF to investigate these processes thoroughly, though such an investigation is outside the scope of this work. This is a given that even on a C-catalyst basis, such work detailing the formation and healing of defects and modelling of the full growth process without employing bias has not been achieved.

We have added the following paragraph to the main text to highlight the role of hydrogen on page 8.

"Note that here, carbon atoms are supplied directly inside the catalyst, which then diffuse rapidly to the surface, rather than via hydrocarbon (CH_x) decomposition. Previous MD simulations³⁹ have shown that CH_x undergoes C-H bond cleavage on the catalyst to form Fe-H bonds which effectively deposits C on the surface, a process with low energy barriers⁴⁰. While the presence of hydrogen may passivate dangling carbon bonds and slow nucleation in the early stages of growth, hydrogen ultimately desorbs from the surface in favor of saturating the catalyst with carbon."

Remark 2.2

-The formation interval δt and lifetime τ of defect both span several orders in the time scale. By analyzing the probability density function, the authors think the defect formation and healing are both stochastic process. However, there is lack of the understanding of a physical picture behind. It is also possible that there is a distribution of defect-catalyst-interface configuration that present different time scales of formation and healing. Is it possible to classify the defect-catalyst interface. The authors should provide more evidence to evaluate the defect formation and healing.

Response 2.2

We respectfully disagree with the reviewer's remark "the authors think the defect formation and healing are both stochastic process". In the manuscript we clearly show that the time between the formation of unique interface defects follows an exponential distribution (see Figure 1e). This distribution describes the time between events in a Poisson process which show that defects form independently of each other and thus the number of interface defects formed during a fixed interval of time is a random variable with a Poisson distribution. In practice, this means that it is not possible to predict exactly when an interface defect will form based on the historical record of interface defect formation.

Likewise, we show that the lifetime of the interface defects follows a power law distribution (with an exponential cutoff). While the power law distribution implies a more complex process with multiple underlying mechanisms compared to the exponential distribution, the lifetimes are still stochastic. That is, when an interface defect is formed, it is not possible to predict exactly how long it will live. Simply put, the formation of interface defects can be considered a single-barrier process, while healing is a more complex multi-step process with different barriers for each step.

To better convey this to the reader we have added the following two sentences to the main text on line 170 and 184 respective.

“The exponential distribution describes the time between events in a Poisson point process, which means that δt is stochastic, i.e. the formation of interface defects follows a simple single-barrier process.”

“The power-law distribution describing τ implies that healing of interface defects is a more complex process than formation involving multiple steps with individual barriers resulting in stochastic lifetimes.”

The reviewer’s remark “It is also possible that there is a distribution of defect-catalyst-interface configuration that present different time scales of formation and healing” is interesting. As stated in the manuscript, we have analyzed the configurations of the SWCNT-edge just before the formation of an interface defect (see Figure 3cii) and compared it to all configurations observed during the entire growth process (see Figure 3ci). Here it is clear that no set of edge configuration stands out. Thus, there are no “special” edge configurations which lead to the formation of interface defects which in practice means that there is no way to predict whether an interface defect will form based on the configuration of the SWCNT-edge.

However, there may exist, as the reviewer points out, a correlation between the configuration of interface defects i.e., the structure of the interface defect as it forms and its corresponding lifetime. We thank the reviewer for pointing this out. To investigate this, we randomly selected interface defects with lifetimes varying by orders of magnitude and compared their structures. Given that the lifetimes span several orders of magnitude any correlation between the configuration of interface defect and its lifetime should be clear. Figure R3 shows the configuration of several interface defects (at the time of formation) grouped by the magnitude of their lifetime, where for each group, five randomly selected interface defects are shown. Comparing the configurations shown in Figure R3a (lifetimes on the order of 10^{-12} s) with that of Figure R3d (lifetimes on the order of 10^{-9} s) no obvious difference can be seen. We conclude that the configuration of the interface defect at the time of formation does not determine its lifetime.

We have added Figure R3 to the manuscript as Extended Data Fig. 5 along with a corresponding paragraph in the main text on page 16.

“By comparing the edge configurations present just before the formation of interface defects, Fig. 4cii, to those of all the edges seen during growth, Fig. 4ci, it is apparent that formation of interface defects does not depend on the configuration of the edge but is instead purely stochastic. Similarly, there is no apparent correlation between the configuration of interface defects, at the time of formation, and their lifetimes as shown in Extended Data Fig. 5. Thus, the configuration of the interface defect, at the time of formation, does not determine how it heals, resulting in stochastic lifetimes.”

Figure R3. The structure of several randomly selected interface defects (highlighted in blue) at the time of their formation grouped by the order of magnitude of their lifetimes, **a** 10^{-12} s, **b** 10^{-11} s, **c** 10^{-10} s and **d** 10^{-9} s.

We disagree with the reviewer's comment that "there is lack of the understanding of a physical picture behind [formation and healing of interface defects]". Here, the distributions modeling the formation and lifetime of interface defects gives information about the different physical mechanisms involved. As mentioned above for the formation of interface defects, the exponential distribution implies that a simple single-barrier mechanism is involved. It's clear in Extended Data Video 2 and 3 that this mechanism is the closure of an open ring (formation of C-C bond) at the edge.

For the interface defect lifetimes, the power law distribution implies that complex multistep mechanisms are involved with varying energy barriers both in number and magnitude. To precisely describe and characterize every mechanism that may be involved in the healing process would require following the trajectory of several thousand of interface defects as they heal, a daunting task that is beyond the scope of this work. We can however from Figure 2d and Extended Data Video 2, 3 derive some key mechanisms for the healing of interface defects.

1. Etching of the SWCNT-edge: The removal of carbon atoms from the edge (etching) of the tube is key to exposing the interface defect to the catalyst where it can heal.
2. Carbon-carbon bond cleavage: Opening of the ring which forms the interface defect, whether pentagon or heptagon, is essential to heal the defect. This, like etching, requires cleavage of carbon-carbon bonds at the edge of the tube.
3. Stabilization of open rings: There are two ways of healing an interface defect; removing it entirely (etching), or converting it to a hexagon. The latter either requires that open rings are held open long enough so that additional carbon atoms can be added (pentagons->hexagons), or reconfiguration of the edge by the conversion of the heptagon to hexagon as seen in Figure 2dii.

Our findings on the key mechanisms of interfacial defect healing provide important physical understanding of defect-free growth and can, for example, guide the choice of catalyst as a solid vs. a liquid catalyst may affect the etching of the CNT edge.

To clarify the key mechanisms involved in healing of interface defects we have extended the description of them on page 9.

"1. Etching of the SWCNT-edge: The removal of carbon atoms from the edge (etching) of the tube is key to exposing the interface defect to the catalyst where it can heal.

2. Carbon-carbon bond cleavage: Opening of the ring which forms the interface defect, whether pentagon or heptagon, is essential to heal the defect. This, like etching, requires cleavage of carbon-carbon bonds at the edge of the tube.

3. Stabilization of open rings: There are two ways of healing an interface defect; removing it entirely (etching), or converting it to a hexagon. The latter either requires that open rings are held open long enough so that additional carbon atoms can be added (pentagons → hexagons), or reconfiguration of the edge by the conversion of the heptagon to hexagon as seen in Figure 2dii.”

Remark 2.3

-According to the manuscript, the defect lifetime is largely independent of the growth rate or carbon supply. But if we consider a faster growth rate (like 5 times higher growth rate), it means the driving force by the Carbon-atom chemical potential between the Fe-C and CNT lattice is higher (5 times higher). In intuition, the large driving force should significantly suppress the etching of CNT. In this view, the defect lifetime should be largely affected by the growth rate.

Response 2.3

This comment is similar to Remark 1.3 from the first reviewer, so we refer to our response there (Response 1.3).

We would like to add here that one must be careful to distinguish between trapped defects (defects inside the tube wall) and interface defects (defects at the tube-catalyst interface). The lifetime, τ , that we consider here is the lifetime of defects at the tube-catalyst interface, i.e., not defects trapped inside the tube wall which by definition have an “infinite” lifetime. If we consider, as the reviewer suggests, a growth rate 5 times that was used to grow the (6,5) tube, we get $k = 2.5 \text{ ns}^{-1}$. This means that, on average, one C atom is added to the growing tube every 400 ps. At this growth rate, our simulations show that defect-free growth is not possible and from such simulations no meaningful conclusions can be drawn about the lifetime of interface defects since most get trapped before they can heal.

Although we agree with the reviewer that at extreme growth rates, where defect-free growth is not possible, the large driving force can suppress etching of the CNT-edge, which may affect the lifetime of interface defects. This is not the regime we are interested in for defect-free growth thus it is out of scope. However, we thank the reviewer for this comment as it raised an interesting point. Our results clearly show that at the growth rates used in our simulations ($k = 0.5 \text{ ns}^{-1}$) we are close enough to equilibrium that the etching of the CNT-edge is not significantly affected.

We have added a sentence on line 201 to emphasize this “Thus, at these growth rates, the system is close to equilibrium and the etching of the CNT edge is not significantly affected, enabling defect-free growth.”

Remark 2.4

-Fig. 3a quite deviates from the experiments. The fastest reported growth rate is about 100 $\mu\text{m/s}$. According to Fig. 3a, all the growth of CNT should be defect free. The similar with the last question, is it possible the defect lifetime is undervalued?

Response 2.4

We thank the reviewer for pointing this out. It is true that Fig. 3a deviates from experiment, however the expected length and thus the map in Fig. 3a, seeks only to qualitatively show how growth rate and temperature affect defect-free growth and is not quantitatively comparable to experiment.

We recognize that this was not clearly stated in the manuscript, and we have now addressed this.

Regarding the reviewer’s question whether the interface defect lifetime may be underestimated, we would like to note the following.

To the best of the authors' knowledge, experimentally reported growth rates are always measured as an average over "long" time periods (seconds). During this period hundreds of thousands to millions of interface defects form and heal. Thus, any fluctuations in temperature during this time will affect the rate of formation and more significantly the rate of healing of interface defects. Likewise, any fluctuation in the carbon supply rate during this time will affect the growth rate and thus impact the likelihood of interface defects getting trapped inside the tube wall.

This lack of temporal resolution in experiments makes it challenging to directly compare the average growth rates measured in experiments with the growth rates measured during MD. This may partly explain why experimentally grown tubes synthesized at significantly lower growth rates compared to our simulations still show many trapped defects. Even though the average growth rate may be very low ($\mu\text{m/s}$), it might fluctuate significantly and thus sometimes be high enough to trap interface defects before they can heal.

Given our careful measurement of interface defect lifetimes over both high and low growth rates and at different temperatures, see Extended Data Table 1, we are confident that our results are correct and that the interface defect lifetime is not underestimated.

We have updated the caption of Fig. 3a which now reads. *"Fig. 3: Influence of growth rate and temperature on defect-free CNT growth. The 2D map in a shows the expected length that a CNT can grow before an interface defect is trapped. To give a better qualitative understanding of the expected length, the value given by Eq. (3) is converted to meters through multiplication by the length per carbon atom of a (11,3) SWCNT ($8.35 \cdot 10^{-12}$ m per C atom). Here the gold star marks the growth conditions used to grow the (6,5) SWCNT in Fig. 2. The plot in b shows the quality of CNTs grown under different experimental conditions, T and P , as determined by the ratio of G-band, I_G , and D-band, I_D , Raman intensities. Here the markers are reproduced from the published experimental data of Picher et al.⁴⁶ and the dashed lines are a linear regression to this data."*

We also revised the main text on lines 213, 221 and 232 to highlight the qualitatively nature of the expected length.

"Thus, a simple qualitative model is proposed for the expected length, in terms of the number of carbon atoms, $\langle N_C \rangle$, that a CNT can reach during growth before an interface defect is likely to be trapped."

"Combined with Eq. (3) it is now possible to construct a qualitative map of the estimated defect-free CNT length for different combinations of growth rates and temperatures."

"These results agree qualitatively with the experimental results of Picher et al.⁴⁴ presented in Fig. 3b where the same trends can be found. Independent experimental results from Vinten et al.⁴⁵ also directly support this."

Remark 2.5

-The author should provide direct simulation result to validate the $\langle N_C \rangle$. The authors could further increase the carbon supply to obtain a shorter $\langle N_C \rangle$ to directly simulate the defect trapping and provide the relation between $\langle N_C \rangle$ and temperature, growth rate.

Response 2.5

While we agree with the reviewer that direct simulation results to validate $\langle N_C \rangle$ would be beneficial, the statistical nature of this metric poses significant challenges. $\langle N_C \rangle$ represents an expected value, therefore a statistical analysis is needed and to collect enough data for this requires very long growth simulations to capture a representative number of interface defects for each run. Additionally, for each combination of k and T , numerous runs are needed to gather a sufficiently large sample size (# added carbon and # of trapped defects) for an accurate calculation of $\langle N_C \rangle$.

We attempted simulations, but the timescales and computational resources/time required for a statistically sound verification of $\langle N_C \rangle$ were prohibitive. Therefore, due to the lack of a sufficiently large sample size, we believe it would be misleading to present these direct simulation results in the manuscript. We have however included our findings in Table R1 for the reviewer's consideration. While the number of total carbons added divided by the total number

of trapped interface defects differ from $\langle N_C \rangle$ the same trends can be seen. As mentioned above in our response to Remark 2.4, the expected length $\langle N_C \rangle$ is intended to provide a qualitative understanding of how growth temperature and growth rate affect the length of defect-free tubes and in this regard, it agrees with our simulations.

Due to the lack of a sufficiently large sample size to perform a statistically sound verification of $\langle N_C \rangle$ we did not add these incomplete results to the manuscript.

Table R1. Direct simulation of $\langle N_C \rangle$. Here, T is the growth temperature, k is the growth rate, and $\langle N_C \rangle$ is the expected length, expressed in the number of carbon atoms. Starting from a snapshot from the growth of the (6,5) tube, 5 runs were performed for each combination of T and k . The third column shows how many of the 5 growth runs failed (due to etching and/or encapsulation). The fourth column shows how many of the successful growth simulations had trapped interface defects. The rest of the columns show the total number of carbon atoms added over all simulations, how many interface defects were trapped, and the total number of carbons added divided by the number of trapped interface defects.

T (K)	k (ns ⁻¹)	# of failed growth runs	# of runs with trapped defects	total # of C added	# of trapped defects	total/trapped	$\langle N_C \rangle$
1000	0.25	0/5	3/5	525	5	105	9871
1000	0.5	0/5	2/5	595	4	149	2552
1000	0.75	0/5	5/5	841	28	30	2178
1250	0.25	0/5	1/5	1399	1	1399	380834
1250	0.5	0/5	1/5	2133	2	1067	6909
1250	0.75	1/5	4/4	1520	14	109	4221
1500	0.25	4/5	0/1	280	0	-	6684951597
1500	0.5	1/5	1/4	1736	2	868	495441
1500	0.75	1/5	1/4	2424	1	2424	153279

Minor remark 2.6

-In the Abstract, "Here, using molecular dynamics simulations driven by a machine learning force field¹³ (MLFF) we developed, DeepCNT-22, " seems not proper. It sounds like the authors developed the MLFE all themselves.

Response 2.6

We thank the reviewer for pointing this out and we have revised the abstract to read.

"Here we present DeepCNT-22, a machine learning force field (MLFF) to drive molecular dynamics simulations through which we unveil the mechanisms of CNT formation, from nucleation to growth including defect formation and healing."

Minor remark 2.7

-In the Abstract, "This contradicts the previous notion of a continuous spiral growth mode¹⁴, but confirms that the growing tube edge exhibits significant configurational entropy¹⁵" is not accurate. The continuous spiral growth mode depends on the practical growth condition. With moderate etching species, the growth is chiral-angle dominant. (ACS Nano 2022, 16, 5627–5635)

Response 2.7

ACS Nano 2022, 16, 5627-5635 presents experimental results showing that chiral angle-dependent growth rates can be achieved under certain conditions. While we do not question the validity of these experimental results, we disagree with the reviewer that this is evidence of continuous spiral growth as other models (for example, ACS Nano 2023, 17, 8, 7135-7144) can also explain these results.

Thus, the experimental results in ACS Nano 2022, 16, 5627-5635 are not direct evidence of continuous spiral growth in contrast to our MD simulations which directly show (at the atomic level) that the tube edge exhibits significant configurational entropy (not continuous spiral growth).

We do however agree with the reviewer that our statement in the abstract is too broad since we have not tested all possible growth conditions thus, *we have revised the statement in the abstract to read “This does not support continuous spiral growth as a general mechanism, as it shows that at these growth conditions the growing tube edge has significant configurational entropy.”*

Minor remark 2.8

-In the Abstract, "healing becomes more efficient than formation, allowing CNTs to grow defect-free to seemingly unlimited lengths" is not accurate. Whether defect trapping or defect free growth should be attribute to the competing of defect healing rate (lifetime) and the lattice growth rate. This sentence is very misleading.

Response 2.8

We thank the reviewer for pointing this out. This sentence does not convey what we intended, and *we have therefore changed it to “We demonstrate that defects form stochastically at the tube-catalyst interface, but under low growth rates and high temperatures, these heal before becoming incorporated in the tube wall, allowing CNTs to grow defect-free to seemingly unlimited lengths.”*

Minor remark 2.9

-Page 7, "From Fig. 2b, it ...", the later claims cannot be directly seen from Fig. 2b.

Response 2.9

We agree with the reviewer that the five distinct phases of growth cannot easily be seen in Fig. 2b, *we have updated the figure to address this, Fig. 2b now looks as follows.*

Here, each phase is denoted by the different shaded regions which are numbered accordingly. The caption of Fig. 2b has also been updated to describe these shaded areas *“The solid black line is the total number of carbon atoms added to the system, the transparent colored lines represent raw data, the solid lines are the result of applying a low-pass filter and the shaded regions denote each phase of growth 1, 2, 3, 4 and 5.”*

Line 121 of main text now reads *“As shown by the different shaded regions in Fig. 2b, growth can be divided into five distinct phases:”*

Minor remark 2.10

-There are quite a few grammatical errors and should be corrected. For example, in line 56, the ' . Which' should be changed into ', which'. In line 71, 'understand for' should be changed into 'understand'. In line 197, 'The map in shown in' should be changed into 'The map shown in'. In Fig. 2 caption, the label C should be lower case.

Response 2.10

We have carefully revised the manuscript and to reduce any grammatical errors and improve the language.

Reviewer #3 (Remarks to the Author):

Response 3.0

We thank the reviewer for co-reviewing this manuscript.

REVIEWER COMMENTS

Reviewer #1 (Remarks to the Author):

The authors have answered all my comments and questions adequately, and I have no further questions or remarks. As such, I recommend the manuscript in its current state for publication.

Reviewer #2 (Remarks to the Author):

Thank the authors for carefully answering my questions and concerns. I still have some questions. If they can properly address them, I would suggest this paper to be published.

1. As for the effect of hydrogen, I understand the explanation about carbon supply regarding to C-H and H-Fe band. However, what I really concern is the hydrogen effect on the C-Fe bond at the interface. At least, the authors should provide an estimation about this effect. How large portion of contributions it could have.
2. I note that the authors mentioned that when an interface defect is formed, its lifetime is stochastic and it is not possible to predict exactly how long it will live. If there is no correlation between the formation time and lifetime for a specific defect, will it be possible to have defects with ultra-fast formation time while having ultra-long lifetime, which might not be healed? According to Fig. 2e and e, this is possible as both the formation time and lifetime span several orders and they have overlaps. In other words, is it fair to compare just the expectation value of defect formation time and defect lifetime?
3. The authors mentioned that the experimentally grown tubes at significantly lower growth rate still show many trapped defects. They mentioned that growth rate might fluctuate significantly in real experiments and sometimes be high enough to trap defects before they can heal. Can the authors comment in what situation can their findings be useful to experimentalists? Is it possible to have a constantly low growth rate for the proposed defect-free growth, especially considering such a high growth temperature is needed?

Reviewer #3 (Remarks to the Author):

We sincerely appreciate the reviewers for their additional comments on our manuscript. Overall, we believe that our responses and revisions comprehensively address the concerns raised by the reviewers. We have highlighted all changes in the manuscript text file using color highlighting for ease of reference.

Below are our point-by-point responses to each of the reviewers' comments structured as follows.

Remark X.Y

Comment of the reviewer.

Response X.Y

Response of the authors.

Action taken when applicable.

Reviewers' comments:

Reviewer #1 (Remarks to the Author):

The authors have answered all my comments and questions adequately, and I have no further questions or remarks. As such, I recommend the manuscript in its current state for publication.

Response 1

We thank the reviewer for taking the time to review our work.

Reviewer #2 (Remarks to the Author):

Thank the authors for carefully answering my questions and concerns. I still have some questions. If they can properly address them, I would suggest this paper to be published.

Remark 2.1

As for the effect of hydrogen, I understand the explanation about carbon supply regarding to C-H and H-Fe band. However, what I really concern is the hydrogen effect on the C-Fe bond at the interface. At least, the authors should provide an estimation about this effect. How large portion of contributions it could have.

Response 2.1

We thank the reviewer for their clarity on their question and respond on multiple fronts. The role of hydrogen in the nucleation and growth of CNTs and the effect of H on the CNT-catalyst interface are longstanding and outstanding research topics in the field. First, extending our current machine learning force field (MLFF) to include H is a non-trivial ongoing part of future research that represents a significant undertaking and requires extensive additional work, so we consider it beyond the scope of this work.

Secondly, in addition to the prior work we mentioned in our previous response, we refer to previously published density functional tight binding (DFTB) molecular dynamics (MD) simulations of CH_x ($x=0, 1, 2, 3$) deposition on Fe_{38} and Ni_{38} catalysts (Eveleens, Page, J. Phys. Chem. C, 2019, 123, 10622-10629). These demonstrate that on Ni, CH_x undergoes dehydrogenation to form C and the released H on Ni then leaves the surface rapidly as H or H_2 . On Fe, there is less C-H bond cleavage than on Ni, though when it occurs, the comparatively stronger Fe-H bond sees a small amount of H remaining adsorbed on the catalyst surface on the simulated timescales (500 ps, at 1500 K). Importantly, regarding the specific query of the reviewer: the surface-bound H does not interfere with the facilitation of chain growth and nucleation into rings, though these rings can stand perpendicular to the surface when fully H-passivated, compared to more-readily surface-adsorbed graphitic networks of pure C. Additionally on Fe, the Fe-C bonds do not grow sufficiently weak that carbon chains dissociate from the surface, despite in some

instances being only bound by single Fe-C bonds. Ultimately, nucleation of graphitic networks and CNT-cap structures still proceeds in the presence of H, albeit slower, with the ratio of defective rings to hexagonal rings decreased compared to growth in the absence of H. If H is present on the catalyst, it does not reside at the Fe-C interface and does not significantly impact the Fe-C bond, as we demonstrate below.

Lastly, we performed additional DFT calculations, which we have added as Section 5 of the Supporting Information, to determine the influence of adsorbed hydrogen on the Fe-C bonds at the interface. We calculated the adhesion energy of (6,6) and (10,0) CNTs attached to a Fe₅₅ nanoparticle with varying degrees of surface-adsorbed H, as shown in Figure R1. Importantly, during relaxation the H atoms initially placed close to the tube-catalyst interface moves to the hollow sites of the Fe nanoparticle as shown in Figure R1a. Furthermore, the adsorbed H has only a marginal effect on the adhesion energy (C-Fe bond strength), regardless of how much surface-bound H is present, or where it is adsorbed on the catalyst, as shown in Figure R1a,b.

We conclude that the presence of H on the catalyst does not significantly affect the Fe-C bond strength and that the Fe-facilitated healing of interfacial defects that we observe in our MD simulations is unlikely to be affected by the presence of surface adsorbed H atoms.

Figure R1. The impact of adsorbed hydrogen on the carbon-metal adhesion energies for (6,6) and (10,0) SWCNTs. **a** the structure before and after relaxation for the high concentration, 37 H atoms adsorbed on the Fe₅₅ catalyst surface. **b** adhesion energies for different concentrations, here hydrogen is adsorbed at random positions on the catalyst surface. The dashed lines are linear regressions to the data points. **c** variation in adhesion energies with respect to the position of the adsorbed hydrogen.

We have added a section to the Supporting Information (see Section 5) detailing the results of our additional DFT calculations. This includes the addition of Figure R1 as Fig. S6 in the Supporting Information. Furthermore, we refer to these additional results in the main text when discussing the effects of hydrogen.

"Note that here carbon atoms are supplied directly inside the catalyst, which then diffuse rapidly to the surface, rather than via hydrocarbon (CH_x) decomposition. Previous MD simulations³⁹ have shown that CH_x undergoes C-H bond cleavage on the catalyst depositing both carbon and hydrogen on the surface, a process with low energy

barriers⁴⁰. However, it has been shown that these surface-bound hydrogens are few in number and not present at the tube-catalyst interface⁴¹. In addition, DFT calculations presented in Section 5 of the Supporting Information show that adsorbed hydrogen on the surface of the catalyst only marginally increases the carbon-metal adhesion energy. Thus, while the presence of hydrogen may passivate dangling carbon bonds and slow down nucleation in the early growth stages, most hydrogen eventually desorbs from the surface and those that remain do not significantly affect the Fe-C bond strength at the interface.”

Remark 2.2

I note that the authors mentioned that when an interface defect is formed, its lifetime is stochastic and it is not possible to predict exactly how long it will live. If there is no correlation between the formation time and lifetime for a specific defect, will it be possible to have defects with ultra-fast formation time while having ultra-long lifetime, which might not be healed? According to Fig. 2e and e, this is possible as both the formation time and lifetime span several orders and they have overlaps. In other words, is it fair to compare just the expectation value of defect formation time and defect lifetime?

Response 2.2

First, we address the question whether there is no correlation between the formation time and lifetime for a specific interface defect. While this is briefly mentioned in the Supporting Information when discussing the derivation of Eq. (3) (Section 4) it is clear from Figure R2 that the time between formation, δt , and the lifetime, τ , of an interface defect is uncorrelated, with a Pearson correlation coefficient of 0.09. Thus, it is indeed possible to have an interface defect with short formation time and long lifetime, however whether the defect will be trapped or not is still only determined by its lifetime (and the growth rate), not its formation time.

Regarding the reviewers’ question whether it is fair to compare the expectation value of the interface defect formation time and defect lifetime the answer is yes. This is because we are interested in modeling the most likely outcome of growth. So, to determine how many interface defects must be created on average before one gets trapped, it is most appropriate to use the expected (mean) value of their lifetime. Then to determine how long the tube is expected to grow before this occurs it is appropriate to use the expected (mean) value of the time between formation of interface defects in combination with the growth rate.

Figure R2. The correlation between the time between interface defect formation, δt , and the interface defect lifetime, τ . Here the blue dots represent an individual interface defect (one value of δt and τ given for each interface defect) and the black lines show their respective probability density functions Eq. (1) and (2) in the main text.

To make it clear that δt and τ are uncorrelated we have added Figure R2 to the Supporting Information as Fig. S5. The text in Section 4 (page 5) of the Supporting Information now reads “As shown in Fig. S5, the lifetime of interface defects, τ , does not depend on the interval between their formation, δt .”

Remark 2.3

The authors mentioned that the experimentally grown tubes at significantly lower growth rate still show many trapped defects. They mentioned that growth rate might fluctuate significantly in real experiments and sometimes be high enough to trap defects before they can heal. Can the authors comment in what situation can their findings be useful to experimentalists? Is it possible to have a constantly low growth rate for the proposed defect-free growth, especially considering such a high growth temperature is needed?

Response 2.3

Since our calculations using MLFFs reach a high level of accuracy and cover sufficiently long timescales, they show that a direct comparison with recent in situ HRTEM growth observations (see for example Yang, et al., Science Advances, 2022, 8, eabq0794., Yang, et al., Journal of the American Chemical Society, 2019, 141, 5871–5879, Wang, et al., Science Advances, 2022, 8, eabo5686 or Lin, et al., Journal of Catalysis, 2017, 349, 149–155) is possible. This provides atomistic details of CNT growth that are not readily available via experiments and are therefore very useful for experimentalists to better evaluate the results from their growth experiments. Moreover, as shown by our results, MLFFs are accurate and efficient enough to simulate CNT growth and therefore can be an important part in creating a digital twin of growth experiments that can be used to design and optimize growth experiments.

Regarding our findings on how growth rate and temperature affect defect-free CNT growth, this is indeed useful for experimentalists as it is now clear that precise control of growth temperatures and carbon supply is important to achieve growth of long defect-free tubes. Experimentalist can now focus on achieving precise control of growth temperatures during growth, which may justify moving away from the traditional hot-wall CVD synthesis. Likewise, achieving precise control over the growth rate may require new methods of supplying carbon to the catalysts, such as diffusion through liquid metals, which can become a focus for experimentalists.

Our results further indicate that high temperatures can enable the growth of long defect-free tubes at high growth rates. Controlled growth at such high temperatures can be challenging with traditional hot-wall CVD synthesis, so exploring other methods such as arc discharge or laser ablation for fast growth of long defect-free CNTs can be a focus for experimentalists. We believe that it is possible to achieve a stable low growth rate by carefully controlling the supply of carbon atoms to the catalyst by, for example, controlling the partial pressure of the carbon feedstock. At high temperatures, this can be challenging with traditional hot-wall CVD synthesis, so as mentioned above, it may be interesting to consider other methods of supplying carbon atoms to the catalyst.

We have added a sentence to the discussion section highlighting the direction that future experiments may take.

“Achieving such growth, with precise control of growth temperatures and rates, may justify moving away from traditional hot-wall CVD synthesis and innovative new methods of supplying carbon to the catalysts, a direction for future experiments.”

Reviewer #3 (Remarks to the Author):

Response 3.0

We thank the reviewer for co-reviewing this manuscript.

REVIEWERS' COMMENTS

Reviewer #2 (Remarks to the Author):

The authors have successfully addressed all the questions and the manuscript now meets the quality for publishing.

Reviewer #3 (Remarks to the Author):
